# End-to-End Deep Knowledge Tracing by Learning Binary Question Embedding

**Hiromi Nakagawa, Kaoru Nasuno, Yusuke Iwasawa, Katsuya Uenoyama & Yutaka Matsuo**
Department of Engineering, The University of Tokyo
{nakagawa,nasuno,iwasawa,uenoyame,matsuo}@weblab.t.u-tokyo.ac.jp

## ABSTRACT

*Knowledge tracing* is the task to estimate student proficiency, and the recently proposed method called Deep Knowledge Tracing (DKT) shows remarkable performance; however, existing DKT requires human labeling that show the required skills to solve a question, which limits the capacity of the model and application to real-world data. In this paper, we propose an end-to-end DKT model, which does not depend on any human labeling. Regarding the process of translating questions into tags as reducing the question-space dimension by a binary embedding matrix, we introduce a new *Q-Embedding Model*, which learns the matrix to help predict student proficiency, and we also present two techniques to learn a better matrix. Using two datasets, we empirically validated that the proposed method show the same or better performance than the DKT using human-defined tags and have an information-efficient structure. These results show the potential of our proposed method to enhance the applicable scope and effectiveness of DKT, which could help improve the learning experience of students in more diverse environments.

## 1 INTRODUCTION

Recent advancements in computer-assisted learning systems have increased the research of knowledge tracing (Corbett & Anderson (1994)), which estimates student proficiency based on their past exercise performance. Piech et al. (2015) reported that the method called Deep Knowledge Tracing (DKT), which leverages recurrent neural networks (RNN) (Williams & Zipser (1989)), performs significantly better than other methods previously proposed.

However, existing DKT models have an essential problem; they need *skill-tags* predefined by human experts that show the required skills to solve each question. In existing DKT, question-space answers are translated into tag-space answers based on a human-defined rule and input into the DKT model. Such a knowledge tracing method, which implicitly depends on the human labeling, presents several problems. One is that the skill-tag quality affects the model's performance, that is, DKT cannot model student proficiency well if the skill-tags are not well-organized. Another is that DKT cannot be applied to data that have no skill-tags, which is often the case with real-world data.

In this paper, we propose a first end-to-end DKT model, which does not depend on human-predefined skill-tags. Regarding the translation of questions into tags as reducing the question-space dimension by a binary embedding matrix, we introduce a new Q-Embedding Model, which learns the matrix to help predict student proficiency purely from the student question-answer logs only. In addition to the above extension, this paper also presents two techniques to learn a better question-embedding matrix: reconstruction regularization of question-space and tag-space and sparse regularization of the question-embedding matrix. In this study, we empirically validate that effective question-embedding is learnable using two open datasets of math exercise.

The main contributions of this work are as follows: 1)We proposed an end-to-end DKT model, which requires no human labeling. The model enables modeling student proficiency independently of human-defined skill-tag quality and performing knowledge tracing with data that have no human labeling. 2)We proposed two techniques to learn a better question-embedding matrix. Using two datasets, we showed that the techniques make it possible to perform knowledge tracing using the learned tags with the same or better performance compared to the human-predefined skill-tags.

## 2 PROPOSED METHOD: Q-EMBEDDING MODEL

The DKT model is trained to minimize the negative log-likelihood of the observed sequence of a student's responses under the model:

$$L_p \quad = \quad \sum_t l(\mathbf{y}_t^T \tilde{\delta}(\mathbf{q}_{t+1}), \mathbf{a}_{t+1}) \tag{1}$$

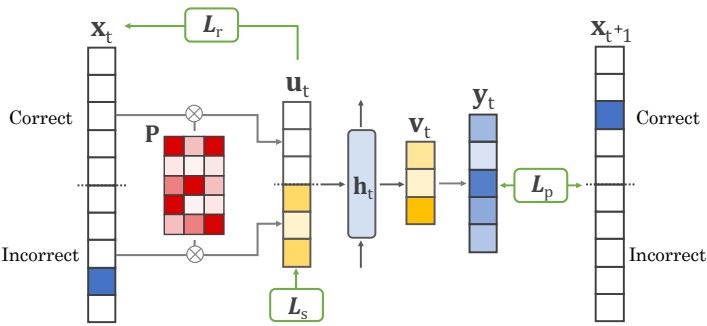

Figure 1: Architecture of Q-Embedding Model

Table 1: Comparison of existing tags and learned tags

| Dataset | Statistics | | | | Results | | | | |
|---|---|---|---|---|---|---|---|---|---|
| | Students | Questions | Skill-tags | Logs | Tags | AUC | Flow hierarchy | GRC | $\sigma$ |
| ASSISTments | 3,410 | 2,635 | 55 | 129,317 | Existing | 0.75 | 0.47 | 0.51 | 3140.94 |
| | | | | | Learned | **0.76** | **0.92** | **0.93** | **1543.96** |
| KDDCup | 1,136 | 3,439 | 192 | 606,819 | Existing | 0.79 | 0.72 | 0.70 | 9701.57 |
| | | | | | Learned | **0.80** | **0.88** | **0.87** | **3674.71** |

where $\mathbf{y}_t$ is the predicted probability of a student answering each exercise correctly; $\tilde{\delta}(\mathbf{q}_{t+1})$ is a one-hot encoding of which exercise is answered at time $t+1$; $\mathbf{a}_{t+1}$ is a vector of whether the exercise is answered correctly or incorrectly (1 or 0) at time $t+1$; and $l$ is the binary cross entropy. However, a student's question-answer logs are not directly used as input, and translated into tag-answer logs based on a human-defined rule and input into the modelx. For datasets with $M$ unique questions and $N$ unique tags, the translation process can be formulated by obtaining tag-ID vector by multiplying a binary matrix $\mathbf{P}$ with a size of $M \times N$ by question-ID one-hot vector, where $\mathbf{P}_{i,j} = 1$ if question $i$ is associated with tag $j$ and $\mathbf{P}_{i,j} = 0$ otherwise. Existing DKT implicitly presupposes this matrix as given; however, the proposed method learns this matrix from students' question-answer logs.

In order to learn the question-embedding matrix $\mathbf{P}$, we introduce the Q-Embedding Model. We present the architecture of the model in Figure 1. In the the Q-Embedding Model, a student's question-answer logs are directly used as the model's input $\mathbf{x}_t$, and the output $\mathbf{y}_t$ is the predicted probability of the student answering each question correctly the next time. In addition, to learn the matrix that translates input question-space to low-dimensional tag-space, we add two hidden layers: $\mathbf{u}_t$ and $\mathbf{v}_t$ with a size of $2N'$ and $N'$, respectively. Here, $N'$ is the dimension of the tag-space and $\mathbf{P}$ is a sigmoid-activated matrix with a size of $M \times N'$. After training the model, we extract $\mathbf{P}$ and binarize it to 0 and 1 on a certain condition.

In addition to the DKT's objective function $L_p$ in equation 1, in order to learn a better question-embedding matrix, we introduce two objective functions in the following equation:

$$L_r \quad = \quad \sum_t l(\mathbf{x}'^T_t \tilde{\delta}(\mathbf{q}_t), \mathbf{a}_t) \tag{2}$$

$$L_s \quad = \quad \sum_t (0.5 - |\mathbf{u}_t - 0.5|) \tag{3}$$

where $\mathbf{x}'_t$ is the vector reconstructed from the first half of $\mathbf{u}_t$ by the same translation as that from $\mathbf{v}_t$ to $\mathbf{y}_t$. $L_r$ is the reconstruction regularization of question-space and tag-space, which aims to reflect the assumption to the training that a student's response to questions is estimable from the student's understanding of each concept corresponding to tag-space. $L_s$ is the sparse regularization, which aims to make $\mathbf{P}$ near 0 or 1 and suppress the information loss when binarizing $\mathbf{P}$ after training the model. Finally, we train the model to minimize the following objective function:

$$L \quad = \quad \alpha L_p + \beta L_r + \gamma L_s \tag{4}$$

where $\alpha$, $\beta$, and $\gamma$ are arbitrary nonnegative real numbers.

## 3 EXPERIMENTS

DKT models student interaction based on the tags associated with the questions; thus, first, we validated the quality of the learned tags by comparing the performance of DKT with the human-defined tags (herinafter called 'existing tags'). we used two open datasets of students' math exercise logs: ASSISTments 'skill_builder'[1] (hereinafter called 'ASSISTments') and Bridge to Algebra

---

[1] https://sites.google.com/site/assistmentsdata/home/assistment-2009-2010-data/skill-builder-data-2009-2010

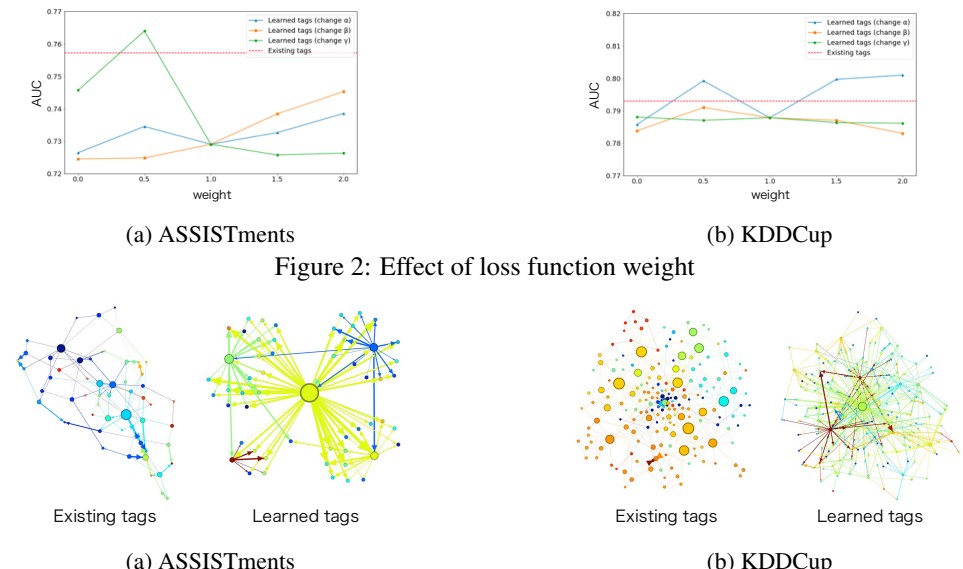

(a) ASSISTments

(b) KDDCup

Figure 2: Effect of loss function weight

| Existing tags | Learned tags | Existing tags | Learned tags |

(a) ASSISTments

(b) KDDCup

Figure 3: Exercise influence graphs

2006-2007 (Stamper et al. (2010)) (hereinafter called 'ASSISTments'). The statistics of the preprocessed datasets are shown in Table 1. We unified the Q-Embedding Model's tag-space dimension with the number of the existing tags: $N' = 55$ in ASSISTments and $N' = 192$ in KDDCup. After training the Q-Embedding Model, we extracted $\mathbf{P}$ and binarized it to $\mathbf{P}'$, where $\mathbf{P}'_{i,j} = 1$ if $\mathbf{P}_{i,j} = \max(\mathbf{P}_i)$ or $\mathbf{P}_{i,j} \geq \theta$ and $\mathbf{P}'_{i,j} = 0$ otherwise, searching the threshold $\theta$ as a hyperparameter. Using this binary question-embedding matrix $\mathbf{P}'$, we translated question-space answers to tag-space answers and applied them to DKT.

We present the AUC scores of DKT in Figure 2 and the best one in Table 1, showing the high score in bold for each dataset. In both datasets, the learned tags recorded the same or higher score than the existing tags and this suggests that the proposed model can learn tags that are as effective for knowledge tracing as the human-defined skill-tags, without using any human labeling. In addition, we can see that $\alpha$, the weight of $L_p$, improved prediction performance in both datasets. Although $\beta$ and $\gamma$, the weights of $L_r$ and $L_s$, improved performance in ASSISTments but had little effect in KDDCup, the settings where any one of the weights is 0 showed poor performance in both datasets; thus, it is considered that each loss function had a good influence on learning a question-embedding matrix to some extent. Our future work is to learn the optimal weights automatically or to incorporate new techniques to train discrete neural networks (Courbariaux et al. (2015); Shayar et al. (2017)).

Next, we constructed exercise influence graphs (Piech et al. (2015)) from the trained DKT model. The graph can be regarded as representing the relationships between knowledge, thus we measured its hierarchy by flow hierarchy (Luo & Magee (2011)) and global reaching centrality (GRC) (Mones et al. (2012)) to investigate the learned tags' characteristics as knowledge representation. We show the graphs in Figure 3 and the comparison of hierarchy in Table 1, showing the high index in bold for each dataset.We can see that the network of the learned tags is more hierarchical than that of the existing tags, which can be suited for representing math knowledge and has a potential to improve student learning efficiency (Block & Airasian (1971); Cohen & Hyman (1979); Abelson (2008)).

Finally, we compared the distribution of the number of times each tag appeared in the answer log, which can directly affect the learning of each unit of DKT. We present the standard deviation $\sigma$ of the distribution in Table 1, showing the low value in bold. We can see that the distribution of the learned tags is less-variant than that of the existing tags. Since the proposed method learns the tags with the optimization of neural networks, it seems that the information of each tag is evenly distributed so that the neural network is easy to model the student interaction with the learned tags.

## 4 CONCLUSION

The experimental results show that the proposed method can learn the effective tags for knowledge tracing, which have different characteristics from those of human-defined tags. Although we need to investigate the proposed method's applicable scope on various datasets and make the obtained tags interpretable by humans in order to truly exploit the method in the real educational environments, this study is the first attempt to perform knowledge tracing in an end-to-end manner without depending on any human labeling. We believe our proposed method could help improve the learning experience of students in more diverse environments.

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
