# OpenReview forum: "End-to-End Deep Knowledge Tracing by Learning Binary Question Embedding"
_ICLR.cc/2018/Workshop — Reject_

### Official Review · AnonReviewer1 · 2018-03-08

**Rating:** 5
**Confidence:** 3

**Review:**

This paper introduces a new Q-Emebedding model for Deep Knowledge Tracking (DKT) to estimate student proficiency. The model automatically learns tag embeddings, instead of relying on human-annotated tags.

My main concern is about the presentation of this paper --- I found it not well-motivated and well-explained enough. For example, the authors could include concrete formulations for their Q-Embedding model, e.g., how are P, u, and v defined an computed. It is a bit difficult for the readers to understand the technical novelty of this model just by looking at Figure 1 and the two loss functions.

---

### Official Review · AnonReviewer3 · 2018-03-09
**Missing many details thus require prior knowledge from the readers to understand the problem and the method**

**Rating:** 3
**Confidence:** 2

**Review:**

This paper works on an end-to-end model for knowledge tracing, where question embeddings are learned, each dimension of which indicates a question tag.

The main concern I have for this paper is that it assumes too many prior knowledge from the readers to understand the problem and the method. For example:

(1) There is no example of what a data sample looks like, which makes it difficult to understand the problem;
(2) There is no motivation for the design of the model architecture;
(3) It is not clear to me how the proposed method performs on the benchmarks. For example, how is each metric in Table 1 computed on the dataset and which one is more important for evaluation; and what is the performance of previous state-of-the-art.

The requirement of prior knowledge makes this paper better suit a workshop specific to the domain instead of a general workshop like ICLR.

---

### Official Review · AnonReviewer2 · 2018-03-12
**cool task but hard to understand the design decisions made**

**Rating:** 4
**Confidence:** 3

**Review:**

This paper addresses the problem of knowledge tracing, in which a model tries to predict a student's future performance based on their past performance. The key difference from past work is that this paper assumes no access to "skill tags", which indicate what skills a particular question needs to be solved. Instead, these "skill tags" are learned by the model. I think this is a cool problem, but the paper is hard to understand; in particular, design decisions such as constraining the learned tag matrix to binary are not properly motivated, and there are no qualitative comparisons to learned tags. Overall, I hope the authors can spend more time explaining why they built their model the way they did in future versions of the paper; as is, I cannot recommend its acceptance.

detailed comments:
- Please give more explanation as to what exactly knowledge tracing is (i.e., with an example showing skill tags etc.). Currently, the introduction is very hard to follow (what does a skill tag look like? what are"question-space" and "tag-space" answers?)
- Please explain Figure 1 in the caption, or at least a high-level overview of what I'm supposed to take away from the figure.
- Why is it desirable to have a binary tag matrix? For interpretability? It seems like there is no reason to enforce this binary constraint since you do not have ground-truth tag labels... If you didn't have the binary constraint, couldn't you just put attention over a real-valued matrix and interpret the attention weights?
- What do you mean by "binarize it to 0 or 1 on a certain condition"? What is this condition?
- Do not understand Figure 3; what am I supposed to learn from it? How can I conclude that the learned tags are "more hierarchical" from these graphs?
- How do the learned tags qualitatively compare to the human ones? This seems like the main motivation of the paper: to show that we can learn tags that are as interpretable as human ones (otherwise why use a binary matrix)?

---

### Public Comment · ~Oriol_Vinyals1 · 2018-02-17
**Please Fix Length**

Your paper violates by a few lines the 3 page limit (see https://iclr.cc/Conferences/2018/CallForWorkshops). Please send us a fixed version of your PDF at iclr2018.programchairs@gmail.com by the end of Monday, February 19th, or else we will reject your paper.

Thanks,
ICLR2018 Program Chairs

---

### Decision · Program_Chairs · 2018-03-20
**ICLR 2018 Workshop Acceptance Decision**

**Decision:**

Reject

**Comment:**

Based on the reviews, this paper has not been accepted for presentation at the ICLR workshop. However, the conversation and updates can continue to appear here on OpenReview.